# Photoprotective Role of Photosynthetic and Non-Photosynthetic Pigments in *Phillyrea latifolia*: Is Their “Antioxidant” Function Prominent in Leaves Exposed to Severe Summer Drought?

**DOI:** 10.3390/ijms22158303

**Published:** 2021-08-02

**Authors:** Antonella Gori, Cecilia Brunetti, Luana Beatriz dos Santos Nascimento, Giovanni Marino, Lucia Guidi, Francesco Ferrini, Mauro Centritto, Alessio Fini, Massimiliano Tattini

**Affiliations:** 1Department of Agriculture, Food, Environment and Forestry (DAGRI), University of Florence, 50019 Sesto Fiorentino, Italy; antonella.gori@unifi.it (A.G.); luanabeatriz.dossantosnascimento@unifi.it (L.B.d.S.N.); francesco.ferrini@unifi.it (F.F.); 2Institute for Sustainable Plant Protection (IPSP), National Research Council of Italy, 50019 Sesto Fiorentino, Italy; cecilia.brunetti@ipsp.cnr.it (C.B.); giovanni.marino@ipsp.cnr.it (G.M.); mauro.centritto@cnr.it (M.C.); 3Department of Agriculture, Food and Environment, University of Pisa, 56124 Pisa, Italy; lucia.guidi@unipi.it; 4Department of Agricultural and Environmental Sciences—Production, Landscape, Agroenergy, University of Milan, 20122 Milan, Italy; alessio.fini@unimi.it

**Keywords:** carotenoids, dihydroxy B-ring-substituted flavonoids, drought stress, epidermal and mesophyll flavonoids, hydroxycinnamates, photoprotection, zeaxanthin

## Abstract

Carotenoids and phenylpropanoids play a dual role of limiting and countering photooxidative stress. We hypothesize that their “antioxidant” function is prominent in plants exposed to summer drought, when climatic conditions exacerbate the light stress. To test this, we conducted a field study on *Phillyrea latifolia*, a Mediterranean evergreen shrub, carrying out daily physiological and biochemical analyses in spring and summer. We also investigated the functional role of the major phenylpropanoids in different leaf tissues. Summer leaves underwent the most severe drought stress concomitantly with a reduction in radiation use efficiency upon being exposed to intense photooxidative stress, particularly during the central hours of the day. In parallel, a significant daily variation in both carotenoids and phenylpropanoids was observed. Our data suggest that the morning-to-midday increase in zeaxanthin derived from the hydroxylation of ß-carotene to sustain non-photochemical quenching and limit lipid peroxidation in thylakoid membranes. We observed substantial spring-to-summer and morning-to-midday increases in quercetin and luteolin derivatives, mostly in the leaf mesophyll. These findings highlight their importance as antioxidants, countering the drought-induced photooxidative stress. We concluded that seasonal and daily changes in photosynthetic and non-photosynthetic pigments may allow *P. latifolia* leaves to avoid irreversible photodamage and to cope successfully with the Mediterranean harsh climate.

## 1. Introduction

The ability of plants to cope successfully with a range of environmental stressors depends on a suite of integrated and modular adjustments involving morphoanatomical, physiological and biochemical traits [1,2,3]. These adjustments are particularly significant for the survival, rather than for the profitable growth, of plants inhabiting highly unfavorable ecosystems, such as the Southern Mediterranean basin [4,5,6]. This concept is consistent with the fact that species evolved in adverse Mediterranean regions, particularly the sclerophyllous evergreens, invest a large portion of the fixed carbon for leaf construction and for the biosynthesis of carbon-based secondary compounds, rather than sustaining new growth [7,8,9]. Mediterranean evergreen species display low CO_2_ assimilation rates (on a leaf area basis), even under the most favorable environmental conditions [4]. Hence, they are daily exposed to an excess of solar irradiance and to the consequent photooxidative stress [10,11].

Excessive light stress may become particularly severe during the central hours of the day, especially in summer, when the scarcity of water availability together with the high temperatures substantially limit the leaf’s ability to use the radiant energy for photosynthesis [4,12,13]. The necessity of coping with an excess of solar irradiance conforms to evergreen sclerophyllous shrubs displaying constitutive morphoanatomical features that are well suited to limiting the deeper penetration of additional photons in the more sensitive leaf tissues [8,14]. Some of these features include leaves that grow at a steep angle, usually with a thick cuticle and mesophyll (at full developmental stages), and covered by a dense indumentum of light-reflecting and absorbing structures (i.e., a wide array of glandular and non-glandular trichomes) [7,8,9]. 

Additionally, effective photoprotection might be provided by specialized secondary metabolites, which may serve the dual function of “avoiding” (limiting) and/or “countering” photooxidative damage [15,16,17,18]. This is the case of xanthophylls, whose biosynthesis is strongly modulated by light [19,20]. For example, violaxanthin cycle pigments (VAZ) are involved in the thermal dissipation of excessive energy through non-photochemical quenching (NPQ), as well as in reducing the oxidative load inside the chloroplasts [15,21,22,23]. Indeed, there is evidence that zeaxanthin may behave as an antioxidant in chloroplasts of leaves exposed for a long time to full sunlight [16,24,25], when the pool of VAZ may saturate the binding sites of the light-harvesting chlorophyll–protein complexes [26,27]. Zeaxanthin may increase the rigidity of the thylakoid membranes, thus reducing peroxidative damage [21,28,29,30]. In addition, it can quench the singlet oxygen produced at considerable rates under drought stress-induced severe excesses of light [22]. In fact, the biosynthesis of zeaxanthin occurs not only through de-epoxidation of violaxanthin but also through hydroxylation of ß-carotene under severe drought stress [25,31,32,33]. 

Pigments biosynthesized through the phenylpropanoid pathway (which is under strict light control [17,34,35,36]), particularly the vast class of flavonoids, may also constitute effective shields against the most energetic solar wavelengths. Additionally, they can act as scavengers of a wide range of reactive oxygen species (ROS), especially when the light stress becomes severe [37,38,39,40]. Indeed, flavonoids are found in very high concentrations in the epidermal cells (due to the small volume in which they are “dissolved”), but they also accumulate to great extent in the mesophyll of leaves adapted or long acclimated to high solar irradiance [14,18,37,41,42]. There is compelling proof that flavonoids located in the vacuoles, chloroplasts and nuclei of mesophyll cells may effectively scavenge the ROS produced during severe light excess [39,40,42]. This dual role of flavonoids in photoprotection (shields and ROS scavengers) is supported by the observation that high solar irradiance, even in the absence of UV radiation, leads to an enhanced biosynthesis of dihydroxy B-ring-substituted flavonoids, whereas it barely affects the biosynthesis of the monohydroxy B-ring-substituted forms (for review articles, see [18,40,42]). Indeed, dihydroxy flavonoids have considerably greater ability to scavenge ROS but very similar UV-absorbing capacity if compared with monohydroxy flavonoids [42].

Mechanisms of photoprotection are activated on a daily basis, as has long been reported for diurnal variations in the concentration and composition of VAZ pigments, thereby sustaining thermal dissipation of excessive radiant energy through NPQ [43,44,45]. Less is known about the seasonal variations in the carotenoid concentration and composition of Mediterranean plants. However, in a range of plant species, evidence of both winter-to-summer and spring-to-summer decreases in the concentration of these molecules (on a leaf mass basis) have been reported [46,47,48,49,50,51]. Much less is known about seasonal and daily variations in phenylpropanoid contents [50,51,52,53,54,55]. Nonetheless, recent findings of significant dawn-to-midday changes in UV-A transmittance, particularly in species growing in warm areas [56,57], suggest that flavonoids other than carotenoids may have diurnal plasticity [58]. Although the extent to which phytochemicals, particularly phenylpropanoids, vary on a daily basis is strongly species-dependent [59], Barnes et al. [56] have provided evidence that diurnal adjustment in ultraviolet sunscreen protection is widespread among higher plants.

Considering this scenario, we hypothesize that both carotenoids and phenylpropanoids may serve a dual role in avoiding and in countering photooxidative stress, and that the relative significance of these two functions is dependent on the severity of drought-induced light excess. For this purpose, we conducted a observational field study on *Phillyrea latifolia*, a Mediterranean evergreen sclerophyllous shrub that displays a strict anisohydric (and water spending) behavior to cope with drought [3]. *P. latifolia* represents an interesting species to elucidate the putative dual photoprotective function of phenylpropanoids, since it possesses a wide array of structures, comprising monohydroxy- and dihydroxy-B-ring-substituted flavones and flavonols [9,14,60]. To test our hypothesis, we carried out both physiological analyses and quantification of individual photosynthetic and non-photosynthetic pigments during spring and summer and at different hours of the day. We also quantified, for the first time, the accumulation of major individual phenylpropanoids, namely caffeic acid and flavonoid derivatives [14,41,42], in different leaf tissue layers.

## 2. Results

### 2.1. Effects of Season and Hour of the Day on Water Relations and Gas Exchange

The physiological traits examined in our study varied considerably on a daily and particularly on a seasonal basis (Table 1, Figure 1). Leaf water potential (Ψ_w_) significantly changed between seasons, with steep spring-to-summer reductions (−85%), irrespective of the hour of the day (Figure 1a). Leaf Ψ_w_ also changed during the day, with substantial morning-to-midday declines (−58%on average), irrespective of the season. Relative water content (RWC) significantly declined from spring to summer (−20%), and daily variations in RWC were more noticeable in summer leaves (Figure 1b). Net photosynthesis (A_N_) decreased (−32%) from spring to summer, irrespective of the hour of the day. In addition, in both seasons, A_N_ declined from early morning to midday (on average −23%), especially in summer (−30%), and fully recovered in the early afternoon (Figure 1c). Overall, summer leaves displayed a lower (on average −35%) capacity to use radiant energy for photosynthesis (iRUE, instantaneous radiation use efficiency, sensu Penuelas et al. [61]) compared with spring leaves (Figure 1d). Greater morning-to-midday reductions in iRUE were also observed in summer leaves (−65% versus −45% of spring leaves).

### 2.2. Effects of Season and Hour of the Day on Photosynthetic and Non-Photosynthetic Pigments

The content of the photosynthetic pigments varied on a seasonal but also on a daily basis (Table 1, Figure 2). The concentration of chlorophyll (Chl_tot_) decreased (−11%) from spring to summer, and from early morning to midday hours, especially in spring leaves (−10%; Figure 2a). However, the ratio of Chl_a_ to Chl_b_ (Chl_a_/Chl_b_) was higher in summer leaves than in spring ones, with a greater increase from morning to midday (Figure 2b). Similar to Chl_tot_, carotenoids (Car_tot_) decreased from spring to summer (−15%) and declined from morning to midday, especially in spring (Figure 2c).

Neoxanthin (data not shown), lutein and ß-carotene changed slightly on both a seasonal and a daily basis (Table 1), although ß-carotene showed significant reductions from early morning to afternoon (12 and 15 h) both in spring and summer leaves (Figure 3a,b). Conversely, violaxanthin cycle pigment relative to the total chlorophyll concentration (VAZ Chl_tot_^−1^) greatly changed depending both on the season and the hour of the day (Figure 3c,d, purple lines). The daily changes in both VAZ Chl_tot_^−1^ and the de-epoxidation state (DES) of VAZ showed different trends in spring and summer leaves, with the values of both parameters being higher in summer (Figure 3c,d). When individually evaluated, it was observed that the concentration of violaxanthin (V) did not change much, whereas antheraxanthin (A) and particularly zeaxanthin (Z) concentrations increased markedly from spring to summer, especially in the afternoon (Figure 3e,f). Violaxanthin and zeaxanthin displayed a reverse daily trend in spring leaves (Figure 3e). In particular, the Z content increased from early morning to noon, followed by a decline in the early and late afternoon. However, in summer, the sharp morning-to-midday increase in Z was not completely matched by a parallel decrease in V content (Figure 3e,f). In addition, Z did not decline from midday to early afternoon (Figure 3f). 

The variation in the content of hydroxycinnamic acid derivatives (HCA_tot_, mostly consisting of echinacoside and verbascoside) and flavonoids (Flav_tot_) was remarkable on botha seasonal and a daily basis (Figure 4a,b, Table 1). The spring-to-summer increase in the flavonoid content (+189%) was mostly due to the increment in quercetin-3-*O*- (+415%) and luteolin-7-*O*-derivatives (+218%), and to the increase in luteolin-4′-*O*-derivatives (+61%), to a minor extent (Figure 4c–f, Table 1). The content of kaempferol and apigenin derivatives, which correspond to approx. 1.0% of the Flav_tot_, slightly varied both seasonally and daily (Table 1, Figure 4c,d, pink lines). Interestingly, the quercetin-3-*O*- and luteolin-7-*O*-derivatives contents showed large morning-to-midday increases (Table 1, Figure 4c,d, black lines; Figure 4e,f, light blue lines). As also observed for Z, the contents of quercetin-3-O and luteolin-7-*O*-derivatives did not decline from midday to early afternoon in summer leaves (Figure 4d,f), whereas they significantly declined over the same time interval in spring leaves (Figure 4c,e).

### 2.3. Tissue Distribution of Hydroxycinnamic Acid Derivatives and Flavonoids in Spring and Summer Leaves

The HPLC-DAD analysis showed that the general phenylpropanoid profile was similar among the different leaf tissues (adaxial and abaxial epidermis, and the adaxial, inner and abaxial mesophyll), with 13 peaks being detected and identified as follows: 1, quercetin derivative; 2, luteolin-7-*O*-Glc derivative; 3, quercetin derivative; 4, luteolin-7-*O*-Glc derivative; 5, hydroxycinnamic acid derivative; 6, kaempferol derivative; 7, apigenin derivative; 8, hydroxycinnamic acid derivative; 9, hydroxycinnamic acid derivative; 10, apigenin derivative; 11, apigenin derivative; 12, luteolin-4′-*O*-Glc derivative; 13, luteolin-7-*O*-Glc derivative (Figure 5). However, despite being similar in composition, the concentrations of these phenylpropanoids proved to be different in each leaf tissue (Figure 5 and Figure 6). In addition, the content of these compounds greatly differed between seasons, being higher in summer (Figure 6).

Epidermal flavonoids corresponded to 30% and 20% of the whole-leaf flavonoid content in spring and summer leaves, respectively (Figure 6a,e). This is consistent with the observation that the mesophyll flavonoids increased to greater extent (+138%, Figure 6b–d) compared with the epidermal ones (+55%, Figure 6a,e) from spring to summer. While the different flavonoids accumulated in the adaxial epidermis almost uniformly, irrespective of the season (Figure 6a), quercetin-3-*O*- and luteolin-7-*O*-derivatives (gray and orange bars, respectively) mostly accumulated in the mesophyll cells (contributing an average of 63% of the whole-leaf flavonoid pool), particularly in summer leaves (69%, right panels). Luteolin-4′-*O*-derivatives (yellow bars) contributed more substantially to the total flavonoids in the adaxial epidermis (23%, Figure 6a), but much less in other leaf tissues (15% of Flav_tot_ of the mesophyll (Figure 6b–d) and 12% of the abaxial epidermis (Figure 6e). Kaempferol and apigenin derivatives (pink bars) mostly occurred in the adaxial epidermis (20% of whole-leaf Flav_tot_, Figure 6a), also with a large content (14%) in abaxial epidermal cells (Figure 6e). These compounds were present at very low concentrations in the mesophyll tissues (5% of whole-leaf Flav_tot_, Figure 6b–d). Finally, caffeic acid derivatives (HCA, blue bars) accumulated poorly in adaxial tissues (Figure 6a,b), whereas they were the predominant phenylpropanoids detected in the abaxial epidermis in comparison with the other phenylpropanoids (Figure 6e).

## 3. Discussion

The data of our study show that *P. latifolia* suffered from water stress not only seasonally but also on a daily basis. Declines in both leaf Ψ_w_ and RWC from early morning to midday resulted in significant reductions in net assimilation rates and, consequently, in even greater declines in instantaneous radiation use efficiency. This was particularly evident during summer, when plants suffered from the combined action of soil water deficit and high temperatures. In other words, *P. latifolia* leaves suffered from excess light stress during the central hours of the day, and the severity of this increased from spring to summer. Here, we reason how seasonal and daily changes in the concentration and composition of photosynthetic and non-photosynthetic pigments may allow *P. latifolia* leaves to avoid irreversible photodamage and hence to cope successfully with multiple environmental pressures associated with the Mediterranean climate. 

Firstly, we observed that chlorophylls declined from spring to summer and from early morning to midday, thereby reducing the light absorption centers. This may have an adaptive value for plants facing high solar radiation under warm and dry climates. [62]. In addition, the higher (+29%) Chl_a_/Chl_b_ ratios observed in summer leaves, particularly during the central hours of the day (+37%), may also have an adaptive value under high light stress conditions on both short- and long-term bases. In fact, higher Chl_a_/Chl_b_ ratios increased the proportion of reaction to light-absorbing centers [16,63,64].

Secondly, the data of our study pointed out the key photoprotective role of carotenoids not only on a daily but also on a seasonal basis. In detail, our findings strongly suggested an antioxidant role of carotenoids in leaves suffering from the most severe drought stress. In particular, we observed that VAZ Chl_tot_^−1^ exceeded 50 mmol mol^−1^. These values are consistent with those of plants growing under full sunlight over the entire growing season observed in previous studies [16,25,26,51,65,66]. This implies that a fraction of VAZ was probably not bound to the light-harvesting chlorophyll–protein complexes and hence was residing in other parts of the thylakoids [25,26,27,28,29]. This unbound pool of VAZ increased greatly during the central hours of the day in summer leaves, mostly due to the enhanced biosynthesis of zeaxanthin (Z). Summer leaves suffered from the severe reduction in iRUE induced by drought stress, being exposed to severe photooxidative stress during the central hours of the day. In addition, the morning-to-midday increase in Z concentration (25 mmmol mol^−1^ Chl_tot_ on average) was not matched by a parallel decrease (6 mmol mol^−1^Chl_tot_on average) in violaxanthin (V) concentration but it was parallel to the decrease in ß-carotene content. We argue that a fraction of the Z synthesized from morning to midday was likely through the hydroxylation of ß-carotene [31,32,33], and hence it was not involved in non-photochemical quenching. However, we cannot exclude that the decrease in ß-carotene contents may have partially resulted from its oxidation by singlet oxygen [22]. Zeaxanthin behaves as a chloroplast antioxidant (sensu Halliwell and Gutteridge [67] and Havaux and Garcia-Plazaola [68]), primarily due to its ability to confer rigidity to the thylakoid membranes [68,69,70] but also due to its capacity to quench singlet oxygen [71,72]. In our study, this may have well limited lipid membrane peroxidation and preserved chloroplasts from irreversible photooxidative damage in summer leaves, when drought stress became particularly severe.

Finally, our study offers novel evidence of the daily and seasonal changes in the biosynthesis of individual flavonoids, as well as of their tissue-specific distribution. The large plasticity in flavonoid biosynthesis observed in both the short- and long-term, as well as in the different leaf tissues, poses new questions that need attention, as outlined below.

The finding of the enhanced biosynthesis of flavonoids from spring to summer, when drought stress increased because of reduced precipitation, is consistent with previous studies [38,51,52,73,74] and is possibly related to the evergreen habit of *P. latifolia* [75]. Indeed, the high flavonoid content in summer may equip the severely drought-stressed evergreen leaves with an effective photoprotective arsenal to cope with the excess light. In fact, our study offers novel evidence that the spring-to-summer increment in flavonoid content (~32 μmol g^−1^ DW) almost exclusively involved 3-*O*-quercetin (18.5 μmol g^−1^ DW) and 7-*O*-luteolin (11.5 μmol g^−1^ DW) derivatives. This led us to hypothesize that these flavonoids, besides having limited the entry of shortwave solar radiation, might also have scavenged the ROS generated by the decreased use of photosynthetic active radiation for photosynthesis due to the severe drought [38,40,76,77]. The finding that the spring-to-summer increases in the content of quercetin and luteolin glycosides mostly regarded the mesophyll tissues adds further support to our idea. Indeed, there is relatively old evidence that dihydroxy B-ring-substituted flavonoids are located not only in the vacuole but also in the chloroplasts of the mesophyll cells in *P. latifolia* adapted to full solar irradiance. Therefore, these compounds would be capable of scavenging H_2_O_2_ and singlet oxygen [14,38,41,78]. In agreement with our findings, Csepregi et al. [58] recently observed a smaller ratio of epidermal flavonoids to those located in the rest of the leaf of *Vitis vinifera* plants growing under non-irrigated field conditions. They suggest that these phenolics, especially quercetin derivatives, acted mostly as ROS scavengers rather than as UV attenuators [58].

The large fluctuation in the content of dihydroxy B-ring-substituted flavonoids on a diurnal basis is intriguing but, at the same time, poses questions that merit deep attention. Dawn-to-midday enhancements in flavonoid amounts, particularly in Que-3-*O*-glucoside, have been previously observed by Barnes et al. [57]. However, in our study, these morning-to-midday increases in the whole-leaf flavonoid content were strong, especially during the summer period. We surmise that this was the result of the concomitant action of multiple environmental stressors during this season and in the daytime, such as high light irradiance, elevated air temperatures and water deficit. All these stressors could enhance the biosynthesis of flavonoids, likely through ROS/redox modulation of the transcription factors involved in regulating flavonoid biosynthesis [79,80,81,82]. Indeed, as reported above, leaves suffered from severe light excess during the central hours of the day (on average, iRUE decreased by 65%), particularly in summer (−70%), and the consequent transient increase in ROS generation may have triggered the biosynthesis of flavonoids with an effective antioxidant capacity [83,84]. In fact, H_2_O_2_, one of the major species of ROS, has been proposed as a conductor of the daily changes in leaf antioxidant defense, acting as a messenger molecule [58]. Thus, the sharp enhancement in quercetin and luteolin derivatives from the early morning to the central hours of the day may have functional antioxidant significance, especially considering that the activities of the key antioxidant enzymes might have substantially declined with the severity of drought, as already reported in a range of species [85,86]. In our study, summer leaves were severely dehydrated at midday (RWC was 68% on average). As such, the leaf temperature (T) may have greatly exceeded that of the air (32.4 °C on average) because of stomatal constraints. This, in turn, may have led to enzyme inactivation [87,88,89], as already reported for the diurnal variations in antioxidant enzyme activity in drought-stressed *Platanus*× *acerifolia* and *Fagus sylvatica* plants [25,88]. Therefore, the mesophyll-located flavonoids, especially quercetin-3-*O*- and luteolin-7-*O*-glucosides, may have complemented the activity of the antioxidant enzymes to avoid irreversible photodamage during the hottest hours of the day [25,90].

Additionally, it is interesting to note that despite having a luteolin aglycone, the luteolin-4′-*O*-derivatives showed a completely distinct diurnal and seasonal pattern, as well as a peculiar localization, when compared with those with the same aglycone but with -7-*O*-glycolization. Lut-4′-*O*-gly showed low diurnal and even seasonal variations (the opposite of Lut-7-*O*-gly derivatives) and contributed more substantially to the total flavonoids in the adaxial epidermis. These results support our hypothesis, since due to its substitution pattern with the sugar being linked to one of the catechol hydroxyls, this flavonoid should contribute less as an antioxidant in the mesophyll cells during the stress period.

The high daily turnover of quercetin and luteolin derivatives observed here poses the question whether the flavonoids have been degraded because of the combined action of high light and elevated leaf T [91] or have instead been oxidized by the excess of ROS (or through electron donation to vacuolar peroxidase)and not fully recycled to their reduced forms [77,92,93]. Addressing this complex issue requires further investigations aimed at exploring changes in both the sub-cellular distribution of ascorbate (and glutathione as well) [94,95] and the activity of mono-dehydroascorbate [93] in leaves facing severe drought stress. 

The photoprotective roles of monohydroxy B-ring-substituted flavones and flavonols, and of caffeic acid derivatives (HCA), are apparently of minor significance. As noted by Gould [96] and by Agati and Tattini [17], monohydroxy B-ring-substituted flavonoids may be constitutive effective shields against UV radiation, whereas HCAs are mostly devoted to absorbing wavelengths over the UV-B portion. This is of high adaptive value against severe solar radiation. The preferential accumulation of HCA in the inner and abaxial mesophyll and in the abaxial epidermis is not surprising. The location of these compounds is consistent with the notion of strong competition between the early and late branch pathways of the general phenylpropanoid metabolism [97,98], with the flavonoid metabolism being favored under the most severe photooxidative stress conditions [90]. We surmise that the preferential accumulation of flavonoids, especially on the adaxial side, might be also related to the adaxial localization of the palisade parenchyma on dorsiventral leaves, which has a denser occurrence of chloroplasts, structures which are greatly exposed to the imbalance of ROS production and ROS scavenging [99].

Finally, we are aware that our analysis of phenylpropanoids targeted to just caffeic acid and flavonoid derivatives might have underestimated the functional roles of other minor phenolics in the responses of *P. latifolia* to drought stress of increasing severity. On the other hand, it has long been known that under the severe light excess imposed by the combined action of high solar irradiance and water deficit, carbon skeletons and energy are almost exclusively devoted to the biosynthesis of flavonoids, particularly flavonols [9,14,17,37,41,42,97]. Nonetheless, a complete analysis of the huge number of phenolics present in leaf tissues merits future investigation using untargeted metabolomic tools.

In conclusion, our investigation offered compelling evidence of the prominent role of photosynthetic and non-photosynthetic pigments in protecting *P. latifolia* leaves against photodamage at both the diurnal and seasonal timescales, thus helping this plant to cope successfully with the environmental pressures associated with the Mediterranean climate.

## 4. Materials and Methods

### 4.1. Plant Material and Growth Conditions

The study was conducted on plants growing on seashore dunes at Grosseto (Tuscany, Italy), with in situ measurements and sampling conducted in 2 seasons: spring (11–12 and 25–26 May) and summer (4–5 and 22–23 July) in 2019. Physiological measurements and collection of biochemical samples were carried out at 4different hours of the day: at 08:00–09:00 (named here as 8 h), 12:00–13:00 (named here as 12 h), 14:30–15:30 (named here as 15 h) and 17:30–18:30 (named here as 18 h) hours. Meteorological data were recorded at the weather station of the Institute of Biometeorology of the National Research Council of Italy, located 15 km away from the experimental site. Average min/max air temperatures (T) in May were 10.7/21.9 °C and the cumulative precipitation was 100 mm; the average min/max temperatures of 14.5/28.8 °C were recorded in June with no precipitation; in July, the average min/max temperatures were 17.7/ 32.4 °C, also with no prior precipitation measurements. Daily global irradiance was, on average, 24.2 MJ m^−2^ in May, 29.7 MJ m^−2^ in June and 31.7 MJ m^−2^ in July. All measurements and sampling were conducted on clear days. 

### 4.2. Water Relations and Gas Exchange Measurements

Relative water content (RWC) and leaf water potential (Ψ_w_) were measured on fully developed leaves (details of the sample size and the experimental plan are reported in Section 4.4). In particular, for RWC determination, leaves were wrapped in parafilm and transferred to the laboratory in a fridge bag to measure leaf fresh weight (FW). They were then hydrated until saturation for 48 h in darkness to determine the turgid weight (TW). The dry weight (DW) of the leaves was obtained after drying them for 48 h at 80 °C. RWC was calculated as follows (Equation (1)):RWC (%) = (FW − DW)/(TW − DW)(1)

Leaf water potential (Ψ_w_) was measured using a Scholander-type pressure chamber (PMS Instruments, Corvallis, OR, USA). Gas exchange measurements were acquired using a LI-6400 portable photosynthesis system (Li-Cor, Lincoln, NE, USA) equipped with a cuvette of 2 cm^2^, operating under ambient CO_2_ and environmental photosynthetic photon flux density. Photosynthesis (*A*_N_) and instantaneous radiation use efficiency (iRUE) were calculated using the LI-6400 software. 

### 4.3. Analysis of Photosynthetic Pigments and Phenylpropanoids

Individual carotenoids and chlorophylls were analyzed by extraction from fresh leaf material (approx. 400 mg) with 2 × 4 mL of pure acetone (added to 0.8 g L^−1^ calcium carbonate). Aliquots (15 µL) were injected into a Perkin Elmer Flexar liquid chromatograph equipped with a quaternary 200Q/410 pump and an LC 200 photodiode array detector (DAD) (all from Perkin Elmer, Bradford, CT, USA). The pigments were separated using a 250 × 4.6 mm Zorbax SB-C18 (5 µm) column (Agilent Italia, Milan, Italy), kept at 28 °C. The column was eluted for 20 min with a linear gradient solvent system at a flow rate of 1.2 mL min^−1^, from 100% CH_3_CN/MeOH (95/5 *v*/*v*, with the addition of 0.05% triethylamine) to 100% MeOH/ethyl acetate (68/32 *v*/*v*). Individual pigments were identified through a comparison of their retention times and UVspectral features with those of authentic standards. Individual carotenoids and chlorophylls were quantified using the calibration curves of authenticated standards from Extrasynthese (Lyon-Nord, Genay, France) and Sigma Aldrich (Milan, Italy).

The whole-leaf concentration of the major phenylpropanoids was analyzed by extraction from fresh leaves (250 mg) with 4 × 4 mL 75% EtOH (pH 2.5, achieved by HCOOH addition). The supernatant was partitioned with 4 × 4 mL of *n*-hexane and reduced to dryness, and the residue was resuspended in 1 mL MeOH/H_2_O (90/10 *v*/*v*). Aliquots (10 µL) were injected in the same HPLC-DAD equipment as reported above. Individual metabolites were separated using a 150 × 4.6 mm (5 µm particle size) Sun Fire column (Waters Italia, Milan, Italy), operating at 30 °C, at a flow rate of 1 mL min^−1^. The mobile phases were: (A) Milli-Q H_2_O (pH 2.5, acidified by H_3_PO_4_ addition)/CH_3_CN (90/10 *v*/*v*) and (B) Milli-Q H_2_O (pH 2.5, acidified by H_3_PO_4_ addition)/CH_3_CN (10/90 *v*/*v*), eluted in a linear gradient program from 100% A to 100% B during a 45 min run. The major phenylpropanoids were identified using the retention times and UV spectral characteristics of authentic standards (all from Extrasynthese, Lyon-Nord, Genay, France). Our analysis identified two derivatives of caffeic acids (echinacoside and verbascoside in the molar ratio of 15–20/80–85, irrespective of the season and hour of the day, referred as to HCA), and the glycosides of five flavonoid aglycones (quercetin, luteolin 7-O, luteolin 4′-O, kaempferol and apigenin). The content of the different phenylpropanoids is reported in μmol g^−1^ DW.

Qualitative and quantitative analyses of caffeic acid and flavonoid derivativeswere also conducted in different tissues of leaves sampled during the central hours of the day (12.00–14:00 h) in both (25–26) May and (22–23) July, using the method of Alenius et al. [100]. Cross-sections were preliminarily observed under a light microscope (Zeiss Axio-Phot, Carl Zeiss, Jena, Germany) to determine the thickness of the adaxial epidermis, the mesophyll parenchyma (palisade and spongy) and the abaxial epidermis. The analyses were conducted on longitudinal 25 mm^2^ leaf sections, obtained by cryostatsectioning (Leica Cryocut 1800, Leica, Wetzlar, Germany, set at −25 °C) from two replicate specimens for each sampling date, each one consisting of 4–5 leaves. Different leaf tissue layers were obtained by cutting pieces at 40 µm depth from the upper surface (adaxial epidermis), then down in three 120 μm steps to obtain the adaxial, inner and abaxial mesophyll, respectively (more details can be observed in Figure 6). The remaining tissue consisted almost exclusively of abaxial epidermis. The samples were immediately placed in a centrifuge tube containing 5 mL of 75% EtOH/H_2_O adjusted to pH 2.5 with HCOOH (the phenylpropanoid extraction solution) and stored at 4 °C until the HPLC-DAD analysis, which was performed as reported above for the whole-leaf phenylpropanoids. The concentration of individual metabolites (in μmol g^−1^ DW) was finally multiplied by the DW of each tissue layer to calculate the tissue’s phenylpropanoid content (in μmol tissue^−1^).

### 4.4. Experimental Design and Statistical Analysis

The experimental design was completely random, performed on eight individuals of *P. latifolia* selected from a group of 30 plants distributed over a 300 m^2^ area, 200 m away from the sea. Each plant was tagged, and fully developed leaves (from the fiftht or sixth nodes, counting from the shoot apex) were sampled for both physiological measurements and biochemical analyses. In particular, for each hour of the day, water relations and gas exchange were measured on two leaves per plant and combined to make an individual replicate (*n* = 8), whereas, for biochemical analyses, four leaves were pooled together from six different plants (*n* = 6). The content of individual phenylpropanoids in different leaf tissues was measured on four plants (*n* = 4), using the same leaves collected for the biochemical analyses at midday. Data were analyzed using a two-way ANOVA (SPSS v.20; IBM, Chicago, IL, USA), with season and hour of the day as factors, and considering their interaction (differences were considered significant at *p* ≤ 0.05). The graphs were constructed using SigmaPlot Systatsoftware (v.12.5, SystatSoftware, Inc., San Jose, CA, USA).

## Figures and Tables

**Figure 1 ijms-22-08303-f001:**
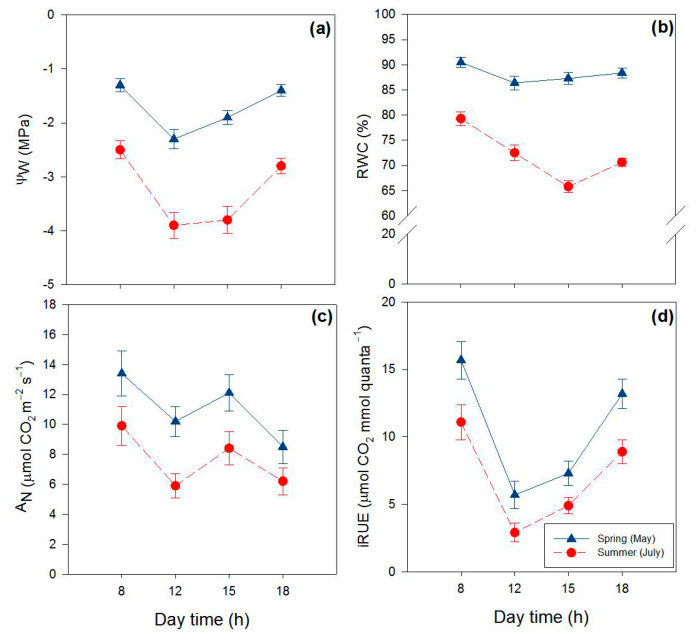
Seasonal and daily variations in leaf water potential (ψ_W_, (**a**)), relative water content (RWC, (**b**)), net CO_2_ assimilation rate (A_N_, (**c**)) and instantaneous radiation use efficiency (iRUE, (**d**)) in *P. latifolia* leaves. Measurements were conducted at 08:00–09:00 (8 h in the graphs), 12:00–13:00 (12 h), 14:30–15:30 (15 h) and 17:30–18:30 (18 h) hours at two dates in spring (11–12 and 25–26 May, blue lines) and summer (4–5 and 22–23 July, red dashed lines) on clear days. Data are reported as means ± SD (*n* = 8).

**Figure 2 ijms-22-08303-f002:**
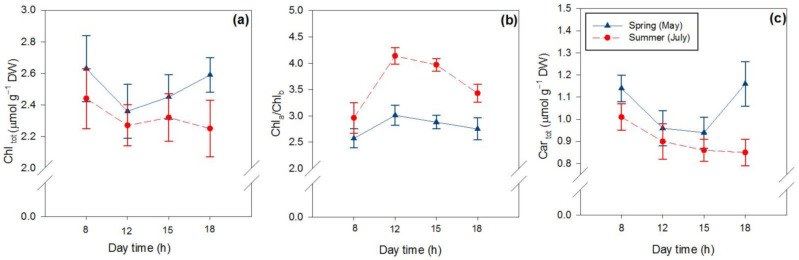
Seasonal and daily variations in total chlorophyll (Chl_tot_, (**a**)), the ratio of Chl_a_ to Chl_b_ (Chl_a_/Chl_b_, (**b**)) and total carotenoids (Car_tot_, (**c**)) in the leaves of *P. latifolia*. Leaves were sampled at 08:00–09:00 (8 h in the graphs), 12:00–13:00 (12 h), 14:30–15:30 (15 h) and 17:30–18:30 (18 h) hours at two dates in spring (11–12 and 25–26 May, blue lines) and summer (4–5 and 22–23 July, red dashed lines) on clear days. Data are reported as means ± SD (* n*= 6).

**Figure 3 ijms-22-08303-f003:**
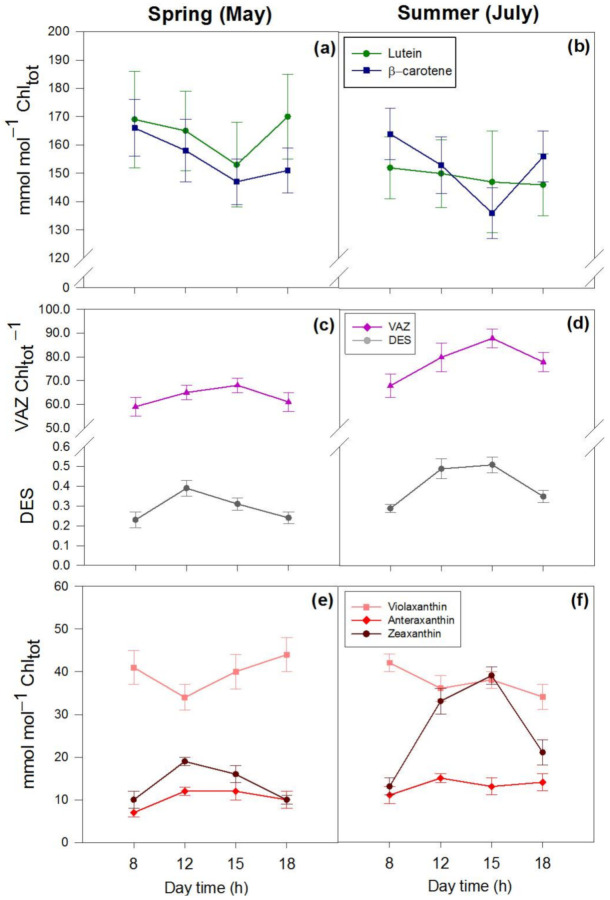
Daily variations in the content of individual carotenoids (mmol mol^−1^Chl_tot_) in spring ((**a**,**c**,**e**)—left) and summer ((**b**,**d**,**f**)—right) leaves of *P. latifolia*. (**a**,**b**) Content of ß-carotene and lutein; (**c**,**d**) ratio of violaxanthin cycle pigments (VAZ) relative to total chlorophyll concentration (VAZ Chl_tot_^−1^) and the de-epoxidation state of VAZ (DES = ((0.5A + V) (VAZ) −1). (**e**,**f**) Content of violaxanthin cycle pigments (violaxanthin, anteraxanthin and zeaxanthin). Leaves were sampled at 08:00–09:00 (8 h in the graphs), 12:00–13:00 (12 h), 14:30–15:30 (15 h) and 17:30–18:30 (18 h) hours ontwo dates in spring (11–12 and 25–26 May) and summer (4–5 and 22–23 July) on clear days. Data are reported as means ± SD (*n* = 6).

**Figure 4 ijms-22-08303-f004:**
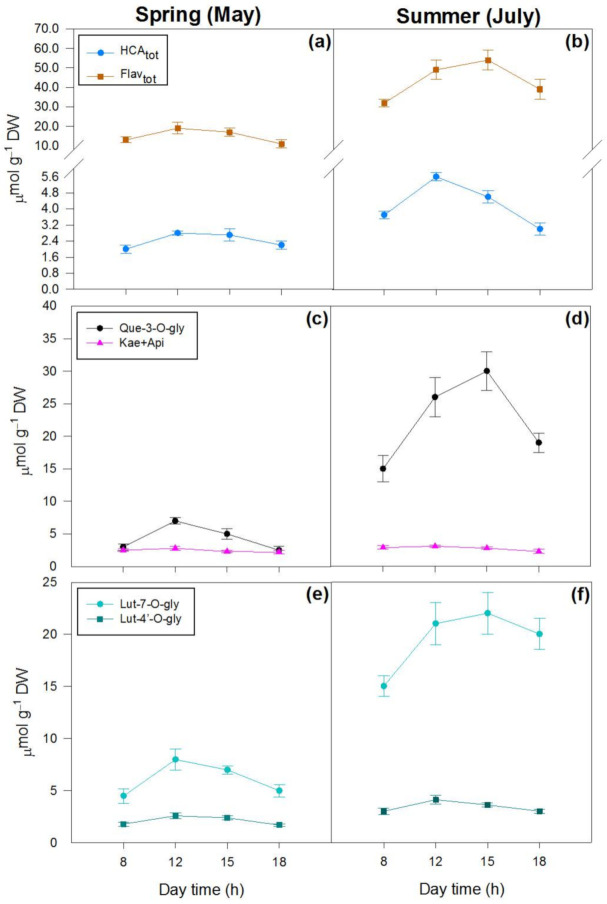
Daily variations in the content (μmol g^−1^ DW) of phenylpropanoids in spring ((**a**,**c**,**e**)—left) and summer ((**b**,**d**,**f**)—right) leaves of *P. latifolia*. (**a**,**b**) Content of hydroxycinnamates (HCA_tot_) and flavonoids (Flav_tot_); (**c**,**d**) content of quercetin-3-*O*-glycosides and kaempferol + apigenin derivatives; (**e**,**f**) content of luteolin-7-*O*- and luteolin-4′-*O*-glycosides. Measurements were conducted at 08:00–09:00 (8 h in graph), 12:00–13:00 (12 h), 14:30–15:30 (15 h) and 17:30–18:30 (18 h) hoursontwo dates in spring (11–12 and 25–26 May) and summer (4–5 and 22–23 July) on clear days. Data are reported as means ± SD (*n* = 6).

**Figure 5 ijms-22-08303-f005:**
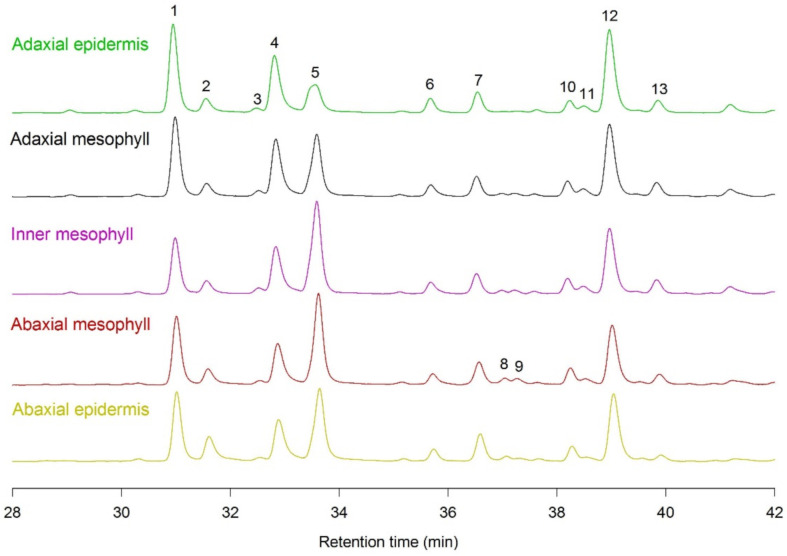
Representative chromatograms (at 310 nm) showing the phenylpropanoid profile of individual tissues (adaxial epidermis—green; adaxial mesophyll—black; inner mesophyll—purple; abaxial mesophyll—red; abaxial epidermis—yellow) of *P. latifolia* leaves sampled during the central hours of the day in summer (22–23 July). Cross-sections were preliminarily observed under light to determine the thickness of the adaxial epidermis, the mesophyll parenchyma (palisade and spongy) and the abaxial epidermis. The analyses were conducted on longitudinal 25 mm^2^ leaf sections, obtained by cryostat sectioning from two replicate specimens, each one consisting of 4–5 leaves. Different leaftissue layers were obtained by cutting pieces at 40 µm depth from the upper surface (adaxial epidermis), then down in three 120 μm steps, to obtain the adaxial, inner and abaxial mesophyll, respectively. The remaining tissue consisted almost exclusively of the abaxial epidermis.

**Figure 6 ijms-22-08303-f006:**
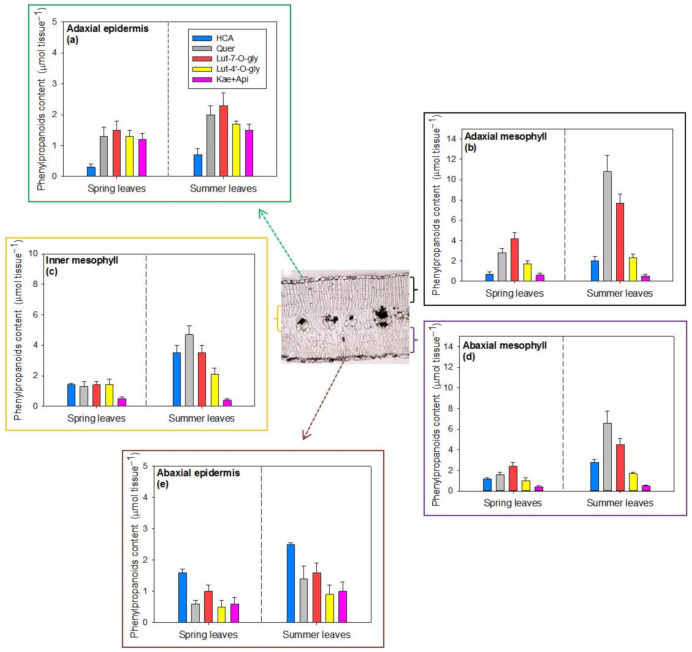
Content of phenylpropanoids (μmol tissue^−1^) in different tissues (different colored rectangles: (**a**), adaxial epidermis; (**b**), adaxial mesophyll; (**c**), inner mesophyll; (**d**), abaxial mesophyll; (**e**), abaxial epidermis) of spring (left-hand bars) and summer (right-hand bars) leaves of *P. latifolia*. Leaves were sampled during the central hours of the day (12:00–14:00 h) in spring (10–11 and 25–26 May) and summer (4–5 and 22–23 July). Analyses were conducted on two replicates for each sampling date, each replicate consisting of 4–5 leaves. The concentration of metabolites (μmol g^−1^ DW) was multiplied by the DW of each tissue layer to calculate the tissue phenylpropanoid content (μmol tissue^−1^). Data are reported as means ± SD (*n* = 4).

**Table 1 ijms-22-08303-t001:** Summary of the two-way analysis of variance (ANOVA; season and time of day as fixed factors, with their interaction factors) for the set of physiological (total degrees of freedom = 63) and biochemical (total degrees of freedom = 47) traits of *Phillyrea latifolia* leaves. In situ measurements and sampling were conducted in two seasons: spring (11–12 and 25–26 May) and summer (4–5 and 22–23 July) in 2019; at four different times of day (08:00–09:00; 12:00–13:00; 14:30–15:30 and 17:30–18:30 h). Ψ_w_, leaf water potential; RWC, relative water content; A_N,_ net photosynthesis; iRUE, instantaneous radiation use efficiency; Chl_tot_, total chlorophyll content; Chl_a_/Chl_b_, ratio of Chl_a_ to Chl_b_; Car_tot_, total carotenoid content; VAZ Chl_tot_^−1^, violaxanthin cycle pigments relative to total chlorophyll concentration; DES, de-epoxidation state of violaxanthin cycle pigments, calculated as (0.5A + Z) (VAZ)^−1^; HCA_tot_, total hydroxycinnamic (mostly caffeic acid) derivatives; Flav_tot_, total flavonoids; Que, quercetin; Lut, luteolin; Kae, Kampferol; Api, apigenin. **** *p* < 0.0001; *** *p* < 0.001; ** *p* < 0.01; * *p* < 0.05; n.s., not significant.

Parameter	F_season_	F_day time_	F_season × hour_
Ψ_w_ (-MPa)	2958.0 ****	385.1 ****	34.9 ****
RWC (%)	1324.5 ****	325.6 ****	147.1 ****
A_N_ (μmol CO_2_ m^−2^ s^−1^)	240.8 ****	23.9 ****	6.5 *
iRUE (μmol CO_2_ mmol^−1^ quanta)	458.6 ****	425.1 ****	2.0 n.s.
Chl_tot_ (μmol g^−1^ DW)	41.1 ****	55.1 ****	27.8 ****
Chl_a_/Chl_b_	212.6 ****	154.2 ****	78.9 ****
Car_tot_ (μmol g^−1^ DW)	54.1 ****	13.2 ***	8.6 **
Neoxanthin (mmol mol^−1^ Chl_tot_)	1.8 n.s.	2.4 n.s.	0.9 n.s.
Violaxanthin (V, mmol mol^−1^ Chl_tot_)	6.2 *	256.8 ****	9.3 **
Antheraxanthin (A, mmol mol^−1^ Chl_tot_)	76.9 ****	285.4 ****	14.8 ***
Zeaxanthin (Z, mmol mol^−1^ Chl_tot_)	455.6 ****	956.0 ****	64.7 ****
Lutein (mmol mol^−1^ Chl_tot_)	9.1 **	5.4 *	2.6 n.s.
ß-carotene (mmol mol^−1^ Chl_tot_)	2.9 n.s.	16.3 **	4.3 *
VAZ Chl_tot_^−1^(mmol mol^−1^)	165.4 ****	38.2 ****	1.5 n.s.
DES	292.6 ****	335.1 ****	25.4 ****
HCA_tot_ (μmol g^−1^ DW)	435.6 ****	84.7 ****	11.5 **
Flav_tot_ (μmol g^−1^ DW)	3953.1 ****	199.7 ****	64.2 ****
Que 3-*O*-der (μmol g^−1^ DW)	3176.3 ***	175.4 ****	89.1 ****
Lut 7-*O*-der (μmol g^−1^ DW)	1412.6 ****	58.6 ****	12.7 **
Lut 4′-*O*-der (μmol g^−1^ DW)	19.3 ***	12.1 **	2.4 n.s.
Kae 3-*O*-der (μmol g^−1^ DW)	6.8 *	1.4 n.s.	1.1 n.s.
Api 7-*O*-der (μmol g^−1^ DW)	5.2 *	1.1 n.s.	0.8 n.s.

## Data Availability

The data presented in this study are available on request from the corresponding author.

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
