# Peer review of "Photoprotective Role of Photosynthetic and Non-Photosynthetic Pigments in Phillyrea latifolia: Is Their “Antioxidant” Function Prominent in Leaves Exposed to Severe Summer Drought?"

_ijms, 2021, doi:10.3390/ijms22158303_

Round 1

Reviewer 1 Report

The manuscript entitled:"Photoprotective role of photosynthetic and non-photosynthetic pigments in P. latifolia: is their “antioxidant” function promi-nent in leaves exposed to severe summer drought?" by Gori et al., describes the phytochemical profile of P.latifolia and its ability to cope with the oxidative stress induced by severe summer drought. Additionally, the authors demonstrated by monitoring some compounds belonging to both carotenoids and phenolics that P. latifolia leaves avoid irreversible photodamages, a typical scenario occurring in the mediterranean harsh climate. 

The manuscript is well written and could be published after a round of minor revision. The major drawback of this work is rapresented by its preliminary and observational nature. In particular, the authors have analysed only few compounds belonging to both carotenoids and phenolics, and this is a severe limitation because they are overlooking the entire pool of phenolic compounds. Just to provide an example, the phenol explorer database consists of more than 700 phenolic compound belonging to at least 5 major classes. Therefore, I'm wondering if the authors have some informations/possibilities to carry out an untargeted phenolic profiling in order to better assess the impact of drought on the phenylpropanoids pathway.

Author Response

Responses to the reviewers

Dear editor,

Please find below a point-by-point reply to the reviewers’ comments.

We are grateful to the reviewers for their useful comments on our manuscript and we hope it will be now suitable for publication in IJMS.

I look forward to hearing from you.

Yours sincerely,

Massimiliano Tattini

Reviewer 1:

The manuscript entitled: "Photoprotective role of photosynthetic and non-photosynthetic pigments in P. latifolia: is their “antioxidant” function prominent in leaves exposed to severe summer drought?" by Gori et al., describes the phytochemical profile of P.latifolia and its ability to cope with the oxidative stress induced by severe summer drought. Additionally, the authors demonstrated by monitoring some compounds belonging to both carotenoids and phenolics that P. latifolia leaves avoid irreversible photodamages, a typical scenario occurring in the mediterranean harsh climate. 

The manuscript is well written and could be published after a round of minor revision. The major drawback of this work is rapresented by its preliminary and observational nature. In particular, the authors have analysed only few compounds belonging to both carotenoids and phenolics, and this is a severe limitation because they are overlooking the entire pool of phenolic compounds. Just to provide an example, the phenol explorer database consists of more than 700 phenolic compound belonging to at least 5 major classes. Therefore, I'm wondering if the authors have some informations/possibilities to carry out an untargeted phenolic profiling in order to better assess the impact of drought on the phenylpropanoids pathway.

We partially agree with the reviewer’s view. She/he is right about the huge number of phenolics that might be involved in the responses of plants to excessive light stress imposed by water deficit. However, only the most concentrated compounds may serve as free radical scavengers and particularly as UV-screening agents (Stafford, 1991; Agati and Tattini 2010). Moreover, our novelty here is the content changes on daily and seasonal basis of individual major compounds, as well as their differential accumulation in the diverse leaf tissues layers. This unlikely is achieved by untargeted metabolomics. We also observe that the flux of carbon and energy is toward the biosynthesis of multifunctional compounds, under light and drought stress, which again cannot involve all phenols, but especially dihydroxy B-ring-substituted flavonoids, as confirmed by our findings. That said, we cannot exclude that the reviewer’s view may merit further investigation. We have therefore, underlined the limits of our study, in the final part of the discussion (lines 392-400). Unfortunately, we do not have possibility to include such relevant untargeted metabolomic analysis due the expressive number of samples harvested in our study. 

Reviewer 2 Report

A well crafted paper and research

there are a few comments but no major concerns 

it is exciting that you have dealt with location of the metabolites   to tissues and even subcellular      

Author Response

Reviewer 2:

A well crafted paper and research there are a few comments but no major concerns  it is exciting that you have dealt with location of the metabolites   to tissues and even subcellular

We thank the reviewer for the comments as well as for the valuable corrections to improve the quality of our manuscript. We revised the whole manuscript as requested by the reviewer. In addition, all the points are systematically addressed below.

Title: should use full name

The correction has been made and the full name of the species was used (line 3).

Abstract: These findings highlight....

The new sentence states: “These findings highlight their importance as antioxidants, countering the drought-induced photooxidative stress” (lines 30-31).

Introduction: leaf's

The correction has been made (line 53).

and being covered

The correction has been made (line 58).

Remove the word “well” from the sentence “of VAZ may saturate well the binding sites”

The correction has been made (line 69).

Table 1: Italic in “In situ”

The correction has been made (line 138).

explain all symbols in legend as well as text

The correction has been made and all the full names have been in the table title (lines 140-144).

Figure 1: think for international audience define what you mean by Summer and Spring total months for the Mediterranean climate. day time in hours?

The correction has been made and the figures 1 to 4 received the specification of the months for each season (Summer – July, and Spring – May) and the unit “h” for the daytime axis. Besides, we better specified the unit of the day time (in hours) in these figure captions. We also corrected the italics in the name of the species in the captions of Figure 2 (line 164) and Figure 3 (line 186).

Discussion: yes was curious about other daily stressor would be helpful to see temperature swings in the leaves  although their vertical arrangement could limit these changes higher temps could also incite ROS production and the pigments could function to alleviate such stress good what about pathogen/insect predation too?

The reviewer is right. Several environmental factors can act as stressors that trigger the production of flavonoids through the ROS production, further providing protection against pathogens and insects attack (so-called cross tolerance).

in the sense of signalling agent? or mechanism to transfer the need for change?

We have completed our sentence with the information “acting as a messenger molecule” in order to clarify how H2O2 acts as a conductor of the daily changes in the leaf antioxidant defense (line 349).

Material and Methods: would it be useful to list soil properties ie soil moisture changes could be influential in results but was there also spiking of samples with the authentic metabolites to examine recoveries?

Unfortunately, we do not have data on soil characteristics. However, our aim was to understand how relevant flavonoids are as antioxidants or as UV-screeners, during light stress of increasing severity. We have previously (long time ago) tested our extraction method to be sure we have fully recovered all the major metabolites. We also checked occasionally the recovery using internal standard.

why  add the term healthy were nearby plants diseased or stressed in other ways some signals from stressed plants are volatiles meaning they could impact neighboring plants

We are grateful to the reviewer for underling this. All the plants used in our experiment were health, as well as the plants nearby. To avoid any misunderstanding, we removed the word “health” from our sentence (line 492).

References: check spaces

            We checked all the mistakes regarding the spaces in the references list. Moreover, we also corrected their numbering and other details. We are grateful for the comment.